# The Impact of the COVID-19 Pandemic on Cancer Patient’s Management—Lithuanian Cancer Center Experience

**DOI:** 10.3390/healthcare9111522

**Published:** 2021-11-09

**Authors:** Daiva Dabkeviciene, Ieva Vincerzevskiene, Vincas Urbonas, Jonas Venius, Audrius Dulskas, Birute Brasiuniene, Ernestas Janulionis, Arvydas Burneckis, Asta Zileviciene, Sigitas Tiskevicius, Rasa Vanseviciute-Petkeviciene, Jurgita Usinskiene, Ruta Briediene, Giedre Bulotiene, Eugenijus Stratilatovas, Valerijus Ostapenko, Jolita Gibaviciene, Ignas Karnas, Saule Kekstaite, Jurgita Navickiene, Albertas Ulys, Algirdas Zalimas, Algimantas Sruogis, Zygimantas Kardelis, Sigitas Zaremba, Renatas Askinis, Saulius Cicenas, Renatas Tikuisis, Ruta Ciurliene, Sonata Jarmalaite

**Affiliations:** 1Biobank, National Cancer Institute, 08406 Vilnius, Lithuania; daiva.dabkeviciene@nvi.lt (D.D.); birute.brasiuniene@nvi.lt (B.B.); 2Cancer Registry, National Cancer Institute, 08406 Vilnius, Lithuania; ieva.vincerzevskiene@nvi.lt; 3Laboratory of Clinical Oncology, National Cancer Institute, 08406 Vilnius, Lithuania; audrius.dulskas@nvi.lt (A.D.); eugenijus.stratilatovas@nvi.lt (E.S.); valerijus.ostapenko@nvi.lt (V.O.); jolita.gibaviciene@nvi.lt (J.G.); albertas.ulys@nvi.lt (A.U.); algis.zalimas@gmail.com (A.Z.); saulius.cicenas@nvi.lt (S.C.); sonata.jarmalaite@nvi.lt (S.J.); 4Department of Medical Oncology, National Cancer Institute, 08406 Vilnius, Lithuania; 5Department of Medical Physics, National Cancer Institute, 08406 Vilnius, Lithuania; jonas.venius@nvi.lt; 6Department of General and Abdominal Surgery, National Cancer Institute, 08406 Vilnius, Lithuania; 7Department of Brachyterapy, National Cancer Institute, 08406 Vilnius, Lithuania; ernestas.janulionis@nvi.lt; 8Department of Radiation Oncology, National Cancer Institute, 08406 Vilnius, Lithuania; arvydas.burneckis@nvi.lt; 9Department of External Beam Radiotherapy, National Cancer Institute, 08406 Vilnius, Lithuania; asta.zileviciene@nvi.lt; 10Department of Nuclear Medicine, National Cancer Institute, 08406 Vilnius, Lithuania; sigitas.tiskevicius@nvi.lt; 11Outpatients Department, National Cancer Institute, 08406 Vilnius, Lithuania; rasa.vanseviciute@nvi.lt (R.V.-P.); giedre.bulotiene@nvi.lt (G.B.); 12Department of Radiology, National Cancer Institute, 08406 Vilnius, Lithuania; jurgita.usinskiene@nvi.lt (J.U.); ruta.briediene@nvi.lt (R.B.); 13Department of Breast Surgery, National Cancer Institute, 08406 Vilnius, Lithuania; 14Department of ENT, Head and Neck Surgery, National Cancer Institute, 08406 Vilnius, Lithuania; ignas.karnas@nvi.lt (I.K.); saule.kekstaite@nvi.lt (S.K.); jurgita.navickiene@nvi.lt (J.N.); 15Department of Oncourology, National Cancer Institute, 08406 Vilnius, Lithuania; algimantas.sruogis@nvi.lt (A.S.); zygimantas.kardelis@nvi.lt (Z.K.); 16Department of Thoracic Surgery, National Cancer Institute, 08406 Vilnius, Lithuania; sigitas.zaremba@nvi.lt (S.Z.); renatas.askinis@nvi.lt (R.A.); 17Department of Anesthesiology and Intensive Care, National Cancer Institute, 08406 Vilnius, Lithuania; renatas.tikuisis@nvi.lt; 18Department of Oncogynecology, National Cancer Institute, 08406 Vilnius, Lithuania; ruta.ciurliene@nvi.lt

**Keywords:** COVID-19 pandemic, cancer management, national lockdown

## Abstract

The pandemic spread of the COVID-19 virus significantly affected daily life, but the highest pressure was piled on the health care system. Our aim was to evaluate an impact of COVID-19 pandemic management measures on cancer services at the National Cancer Institute (NCI) of Lithuania. We assessed the time period from 1 February 2020 to 31 December 2020 and compared it to the same period of 2019. Data for our analysis were extracted from the NCI Hospital Information System (HIS) and the National Health Insurance Fund (NHIF). Contingency table analysis and ANOVA were performed. The COVID-19 pandemic negatively affected the cancer services provided by NCI. Reductions in diagnostic radiology (−16%) and endoscopy (−29%) procedures were accompanied by a decreased number of patients with ongoing medical (−30%), radiation (−6%) or surgical (−10%) treatment. The changes in the number of newly diagnosed cancer patients were dependent on tumor type and disease stage, showing a rise in advanced disease at diagnosis already during the early period of the first lockdown. The extent of out-patient consultations (−14%) and disease follow-up visits (−16%) was also affected by the pandemic, and only referrals to psychological/psychiatric counselling were increased. Additionally, the COVID-19 pandemic had an impact on the structure of cancer services by fostering the application of modified systemic anticancer therapy or hypofractionated radiotherapy. The most dramatic drop occurred in the number of patients participating in cancer prevention programs; the loss was 25% for colon cancer and 62% for breast cancer screening. Marked restriction in access to preventive cancer screening and overall reduction of the whole spectrum of cancer services may negatively affect cancer survival measures in the nearest future.

## 1. Introduction

The unprecedented burden of the causative virus of coronavirus disease 2019 (COVID-19) pandemic on the health system has a huge impact on cancer control worldwide, directly and indirectly affecting all stages of cancer care delivery. In many countries, cancer screening and prevention programs were classified as a low-priority service during the pandemic and were paused or even completely suspended with a significant impact on early detection of cancer [1,2,3]. Cancer treatment pathways were adapted to minimize the risk of infection, and some ongoing services were deprioritized to prevent national healthcare systems from overloading. Moreover, cancer patients in the pandemic appear to be highly vulnerable to worse outcomes due to the disease itself and immunocompromised anticancer treatment [4]. Due to social distancing behavior, many patients were less likely to present to healthcare centers for any non-urgent services [5], and clinicians avoided sending patients to hospitals when possible [6]. The effect of the delay of cancer diagnosis or treatment is not sudden, and changes in a structure of newly diagnosed cancer cases as well as an increase in the number of avoidable deaths might occur in the coming years [7,8].

The first case of COVID-19 infection in Lithuania was confirmed on 28 February 2020 and the first national lockdown in Lithuania was introduced on 18 March 2020 and lasted until 17 June 2020. It caused a dramatic decrease of health services in primary care units and a substantial reduction of cancer screening. Remote consultations took place, and hospitals were accepting only urgent referrals. The decrease of patient flows in specialized (cancer) hospitals was noticeable for a few months and recovered only at the end of the spring of 2020. A second national lockdown was implemented from 4 November 2020 until 1 July 2021 with less marked restrictions in the health sector.

The National Cancer Institute (NCI) is the only cancer treatment-dedicated hospital in Lithuania, providing almost a quarter of national cancer services. From the beginning of the first lockdown in Lithuania, cancer treatment was assigned to high-priority healthcare services, and with strict safety measures was carried out continuously during the whole pandemic. Due to specialization and absence of other clinical departments, the NCI avoided wide infrastructure reorganization and the staff was not resettled to newly established COVID-19 wards, as was the case at other hospitals. Analysis of the changes in patient flow and services might at least partially represent the impact of the COVID-19 pandemic on cancer care in Lithuania. With this aim in mind, the extent and structure of cancer services provided by the NCI during the pandemic period of 2020 was compared to the corresponding period of 2019.

## 2. Materials and Methods

We assessed the time period from 1 February 2020 to 31 December 2020 and compared it to the same period in 2019. For hypofractionated radiotherapy, the time period in 2020 was compared to the same period in 2018, as the data for 2019 were incomplete and fragmented.

In case of prostate cancer patients treated by prostatectomy, the changes were assessed from 1 April to 31 December 2020 vs. 1 April to 31 December 2019.

Data for our analysis were extracted from the NCI Hospital Information System (HIS) and the National Health Insurance Fund (NHIF).

### Statistics

The percent change was calculated, taking the difference between the final value and starting value, dividing by the absolute value of the starting value, and multiplying the result by 100. The percent change was calculated for groups with both starting and final values over 100. Comparison between the number of patients or provided services was performed using Related-Samples Friedman’s Two-Way Analysis of Variance by Ranks (rsANOVA) or the Chi-square test, as appropriate. rsANOVA assessed significant changes in trends between distributions of related samples, while the Chi-square test assessed significant changes between the frequencies of the compared groups. Differences were considered statistically significant when the *p*-value was <0.050. The data were analyzed using R i386 4.0.6 (R Foundation for Statistical Computing, Vienna, Austria); IBM SPSS Statistics 21 (IBM, Armonk, NY, USA).

## 3. Results

### 3.1. Out-Patient Cancer Services

Descriptive statistics for out-patient services at the NCI revealed a marked reduction in 2020 compared to 2019 (rsANOVA *p* = 0.17) according to HIS data. The evident drop in the extent of medical consultations (−14%) and radiology examinations (−16%) or endoscopic (−29%) procedures (Table 1) suggests a deterioration in cancer diagnoses. Additionally, the reduction of follow-up visits (×16%) and the increase in demand of mental health services (+11%) were at least partially related to social distancing measures. The extent of day surgery in out-patient clinics of the NCI was reduced by a quarter (Table 1); this was also a reflection of a reduction in the total amount of surgical services.

The most significant impact of the pandemic was observed for the NCI services related to cancer prevention programs (Table 1). The largest drop of patient flow in 2020 was registered in the breast cancer screening program (−62%), while the lowest impact that the pandemic showed was on colorectal cancer screening (−25%).

### 3.2. The Services of Systemic Anticancer Therapy and Radiotherapy

Systemic anticancer treatment (SACT) is one of the core activities of the NCI with no significant changes in the extent of these services registered during the pandemic (Table 1). Regarding the summarized numbers of out-patient (day care) and in-patient units (Figure 1), time-dependent variations also had no declining trend in 2020 vs. 2019 (rsANOVA *p* = 0.74). However, significant changes in frequency comparing three periods—March–May (−8%), June–August (−1%), and September–December (+9%) (chi2 = 58, df = 2, *p* < 0.001)—revealed that SACT services decreased at the beginning of the quarantine, but later recovered and even increased.

However, the number of unique patients undergoing SACT decreased by 30% (Table 2) in 2020 compared to 2019 (rsANOVA *p* = 0.056), and the changes were tumor localization dependent (chi2 = 1031, df = 6, *p* < 0.001). The most significant increase in number of patients undergoing SACT in 2020 was registered for breast cancer (+38%), while the largest reduction was for gastrointestinal and hepatobiliary cancer (−77%).

Additionally, marked changes in the types of SACT were observed during 2020 vs. 2019, due to the shift to oral medicaments and more efficient targeted anticancer therapies. Overall, in 2020, the extent of chemotherapy procedures decreased by 15%, while biological therapy and immunotherapy increased by 8% and 34%, respectively.

Overall, radiation therapy (RT) provision decreased by 6% in 2020 relative to 2019 (rsANOVA *p* = 0.26, Table 2). No significant tumor localization-dependent changes were noticed (chi2 = 8.1, df = 6, *p* = 0.23). The most significant decrease, by 24%, was detected for gynecological cancer.

The structure of RT services changed towards a more frequent application of hypofractionated RT (HRT). HRT was administered as radical treatment for 29% of patients in 2020, while only 4% of patients received HRT for these purposes in 2018. For palliative care, HRT was also used significantly more often in 2020. In 2018, this treatment accounted for 71% of all patients, and by 2020 it accounted 93%.

### 3.3. In-Patient Cancer Services

To analyze the tendencies in flow of newly diagnosed cases undergoing surgical treatment as a first-choice therapy, several localizations were selected, including colon, stomach, breast, prostate, lung, bladder, thyroid, melanoma, kidney, and larynx cancer. Descriptive statistics revealed trends in a time-dependent decrease of patient flows in 2020 compared to 2019 (rsANOVA *p* = 0.13; Figure 2), and the drops in cases during March–May (−15%), June–August (−12%) and September–December (−5%) were comparable (ch2 = 2.15, df = 2, *p* = 0.34).

Overall, in 2020, the number of newly diagnosed and surgically treated patients decreased up to 10% (Table 3) compared to 2019. These changes were tumor localization dependent (ch2 = 18.2, df = 9, *p* = 0.03; rsANOVA *p* = 0.20). The most significant decrease in the number of operable patients was seen for lung cancer (−36%), while during the pandemic, the extent of chemotherapy/radiotherapy prescriptions for lung cancer patients increased by 16%, suggesting a shift in the stages of newly diagnosed cases. Less changes were observed in surgical treatment of prostate, larynx, thyroid, and kidney cancers as well as melanoma in 2020 vs. 2019.

Overall, a significant decrease in patient number with a clear association to cancer stage was registered in 2020 vs. 2019 (rsANOVA *p* = 0.046, Table 3). However, these changes were unevenly scattered across the tumors of different localizations (Table 3, Figure 3). A reduced number of all-stage patients was characteristic of stomach, breast, or lung cancer (rsANOVA *p* < 0.05) in 2020 vs. 2019, while in lung cancer, over a two-fold reduction of stage I and II lung tumors (ch2 = 7.7, df = 3, *p* = 0.053) was registered. Similarly, the number of cases diagnosed with stage I colon cancer decreased by two-fold, while the extent of stage IV disease increased by 1.4 times (ch2 = 6.6, df = 3, *p* = 0.099) in 2020.

## 4. Discussion

The COVID-19 pandemic and national lockdown negatively affected the Lithuanian healthcare system. According to annual reports and compulsory health insurance database data provided by the Health Information Center of Institute of Hygiene [9], visits to physicians in Lithuania decreased by 34.4%, and visits to oncologists decreased by 20.2% accordingly. The same trend was shown in our study. The pandemic and national lockdown had a negative impact on the majority of cancer diagnostic and treatment-related services provided by the NCI of Lithuania. The number of patients participating in cancer screening programs at the NCI dropped dramatically by more than 25%. Reductions in diagnostic procedures, including radiology (−16%) and endoscopy (−29%), were also observed. The pandemic impacted the extent of out-patient consultations (−14%), disease follow-up visits (−16%), and the numbers of patients undergoing systemic (−30%), radiation (−6%), or surgical (−10%) treatment. Additionally, the COVID-19 pandemic had an impact on the structure of cancer services by fostering the application of biological SACT or HRT. More cancer patients were referred to psychological or psychiatric counselling. The changes of newly diagnosed cancer patients’ flow were dependent on the tumor type and disease stage, showing some rise in advanced disease at diagnosis already during the period of the first lockdown.

Oncology patients belong to the most vulnerable risk groups of COVID-19 infection. Yang and colleagues [9] presented a meta-analysis of 63,019 COVID-19 patients, and 2683 of them were cancer patients. The pooled incidence of cancer in COVID-19 patients was 6% (95% confidence interval, 3–9%), and it was much higher than global cancer incidence (0.2%). Mortality analysis also confirmed a higher risk of death from COVID-19 among patients with cancer diagnosis [9]. In addition, patients with cancer are at an increased risk of more severe infection and subsequent complications, particularly after anticancer treatment [10].

Numerous clinical recommendations, including European multidisciplinary expert consensus [11] for COVID-19 prevention and management, have recently been published to guide cancer patients, health care professionals, and cancer centers on the proper measures for fighting this infection and to maximize the use of the available resources in order to sustain health services for cancer patients. Despite the minor changes in the NCI infrastructure due to the COVID-19 pandemic, strict infection prevention measures were introduced to increase the safety of cancer patients and staff. Frequent staff testing and remote and/or shift work when possible were exploited in order to avoid inside spreading of infection and to prevent departments from a complete shut down due to infection outbreaks. In order to minimize the risk of spreading infection, a shorter patients’ pathway to treatment and reduced number of visits were introduced when possible.

International radiotherapy expert groups published specific recommendations on radiotherapy use in response to COVID-19 [12,13,14], often focusing on the acceptability of hypofractionated concepts in order to spare resources. The same trend was used in the NCI. The irradiation of patients was performed by a strict schedule, separating in- and out-patients with a disinfection interval in between. Thus, the number of patients treated with radiotherapy through the COVID-19 period including in- and out- patients decreased by 6%, and the number of services decreased by 7%. According to the recommendations, treatment schemes for breast, prostate, and lung cancer were modified, reducing the number of fractions and services.

SACT at the NCI was given to the patients according to international guidelines for cancer patient management during the COVID-19 pandemic [15]. In order to decrease potential exposure to COVID-19, SACT modifications were introduced. Intravenous treatment was switched to oral if possible, treatment intervals were changed, and some treatments were postponed. The decrease in patient numbers (−30%) in the Medical Oncology Department of the NCI did not influence the number of provided anticancer treatment services (+4%). These changes can be explained by the introduction of some new targeted therapies and immunotherapies in 2020 with a longer course of treatment. In addition, online consultations were widely used during the pandemic period for patients using oral anticancer drugs.

Several studies [16,17] already noted the trend toward cancer diagnosis at later stages, but this tendency may become more evident in the near future. Evident cancer-specific symptoms, such as a new palpable lump and unusual bleeding, are recognized as alarming signs leading to the medical consultations [18] and probably were not ignored despite the COVID-19 pandemic. Therefore, at the NCI, the number of cancer cases with advanced disease at diagnosis was not impacted by pandemic management measures. However, less specific symptoms, such as weight loss, fatigue, mild pain, and other non-specific worries, could be ignored and visits to healthcare centers postponed or even made impossible due to lockdown regulations [19]. Taking this into account together with the cancer screening programs’ suspension, we can expect patients with more advanced cancer stages in the upcoming years. Such a trend towards advanced disease at diagnosis was already revealed by our analysis of lung cancer cases.

On the country level, the decrease of participation in prevention programs in 2020 was remarkable: −29% for cervical cancer, −43% for prostate cancer, −33% for colorectal cancer, and −28% for the breast screening program according to the NHIF. The number of prevention services at the NCI in 2020 compared to 2019 also decreased. The NHIF data showed that more than 600 cancer cases in Lithuania were not diagnosed in 2020 because of the COVID-19 pandemic’s impact on cancer prevention programs. Considerable increases in cancer deaths can be expected due to delays in diagnostic and screening services; therefore, there is a need to manage a backlog within diagnostic and screening procedures in order to mitigate the impact of the COVID-19 pandemic on cancer patients [19,20].

Today it is quite difficult to predict the complete impact of the COVID-19 pandemic on cancer statistics and the effect possibly will be distinct for different cancer types. Some studies tried to predict the possible influence of diagnosis delays and treatment modifications due to COVID-19 on cancer survival measures. A meta-analysis performed by Hanna et al. shows [7] that even a four-week delay of cancer treatment is associated with increased mortality across surgical, systemic treatmsent, and radiotherapy indications for some cancers, including bladder, breast, colon, rectum, lung, cervix, and head and neck cancers. In case of surgery, this is a 6–8% increase in the risk of death for every four-week delay. The study on management of breast cancer during the pandemic [21] showed that surgery delays of more than 60 days were associated with pathological upstaging in patients with non-invasive ductal carcinoma, but not in those with invasive disease, while no impact of delay in RT up to 6 months for the treatment of prostate cancer patients was found by Dee et al. [22]. Our data and several other studies [16,17,19,20] indicate that undiagnosed cancer cases due to suspending of cancer prevention programs, diagnostics delay, and avoiding visits to health care institutions might be associated with upstaging and a significant rise of newly diagnosed cancer cases with advanced disease.

Furthermore, delayed cancer diagnosis and issues of getting the right and timely treatment have a tremendous impact on the mental health of cancer patients who are at a higher risk of anxiety and depression [23], while the COVID-19-related fear and anxiety increased distress even more [24]. In Lithuania, relatives’ visits to hospitals were mainly prohibited during lockdown with a few exceptions for terminal patients. An increase in numbers of psychological and psychiatric services observed during the pandemic in our study could be related to higher anxiety, feeling of separation from the families, and other psychological issues for cancer patients who faced cancer and the pandemic at the same time [25].

According to the the National Audit Office of Lithuania, during the pandemic, about 10% more people turned to medical institutions for depression, anxiety, and reactions to severe stress disorders, and the number of visits to primary mental health care professionals increased by about 17% compared to 2019.

Our data showed the increase in mental health services for cancer patients. Seeking to overcome hindrances to psychiatric and psychological services due to quarantine restrictions, we adopted telepsychiatry for cancer patients with emergency psychological problems. With the help of remote means of mental healthcare, a larger number of patients were counseled compared to 2019. The techniques of telepsychiatry for cancer patients requires further examination and refinement so that they may be fully utilized within the mental health service system.

Our study has some limitations. First, the analysis is based on single institution data, and the NCI is a specialized cancer center that was less affected by pandemic management measures than clinics involved in COVID-19 care. Secondly, we have compared the lockdown data with only one previous year. Finally, assessment of the long-term effect of the pandemic should be performed and the survival data should be added in the near future.

## 5. Conclusions

Our data showed significant timeframe- and tumor localization-dependent changes in patient and service flows during the COVID-19 pandemic. In order to reduce the negative effect of the pandemic, cancer services should be provided continuously during national lockdowns, ensuring safe COVID-free areas for centralized cancer services. New national strategies for more efficient cancer screening should be prepared in order to compensate the loses in early diagnoses. Future activities need to be done to develop and implement standardized strategies to ensure the psychological wellbeing of patients and medical staff and to improve cancer services during pandemic outbreaks.

## Figures and Tables

**Figure 1 healthcare-09-01522-f001:**
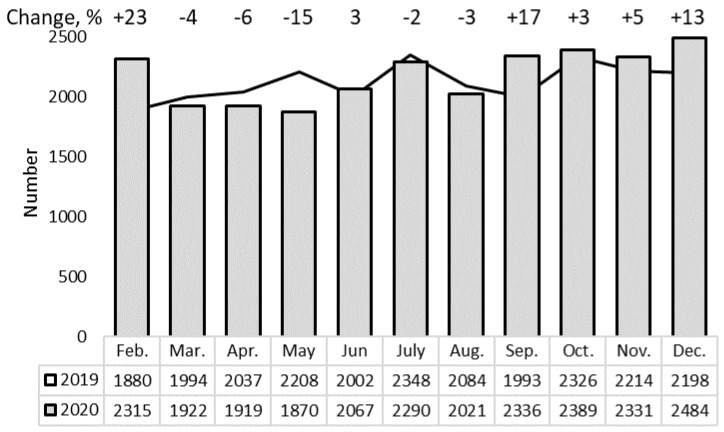
Monthly distribution of systemic anticancer therapy procedures in out-patient and in-patient units.

**Figure 2 healthcare-09-01522-f002:**
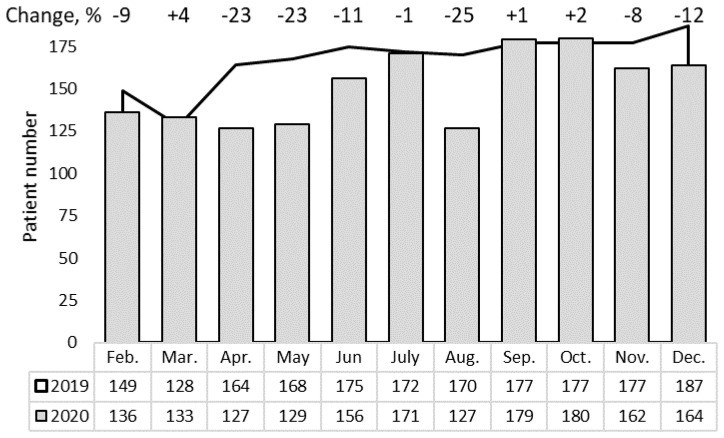
Number of newly diagnosed cancer patients who underwent surgical treatment.

**Figure 3 healthcare-09-01522-f003:**
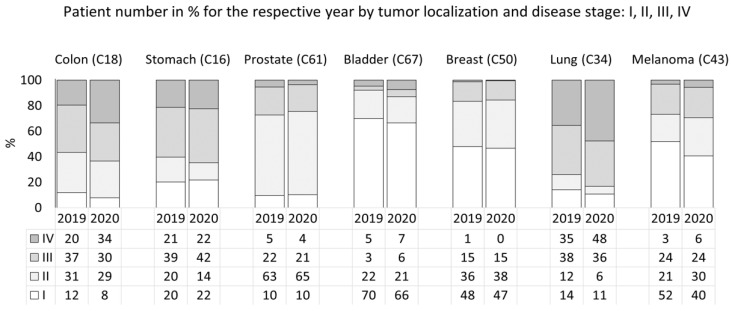
Percentage of surgically treated newly diagnosed cancer patients according to disease stage.

**Table 1 healthcare-09-01522-t001:** Summary of out-patient services at NCI.

Services	2019, (N)	2020 (N)	Change, (%)
Medical consultations	110. 510	94. 709	−14
Mental health (psychological and psychiatric services)	537	597	11
Systemic anticancer therapy	21.623	22.502	4
Radiation therapy	31.854	29.542	−7
Radiology (inc. Rx, MRI, CT, US *)	108.473	90.981	−16
Day surgery	6403	4827	−25
Endoscopic procedures	2627	1853	−29
Disease follow-up visits	16.894	14.205	−16
Preventive programs:			
Mammography	9704	3653	−62
Prostate cancer	259	112	−57
Colonoscopy	780	587	−25

* Rx—X-ray, MRI—magnetic resonance imaging, CT—computed tomography scan, US—ultrasound.

**Table 2 healthcare-09-01522-t002:** Descriptive statistics of patients receiving SACT or radiotherapy by tumor localization.

Solid Tumors	SACT			Radiotherapy/Brachytherapy		
	The Number of Patients		Changes, %	The Number of Patients		Changes, (%)
	2019	2020		2019	2020	
Head-and neck (C00–C14, C30–C32 *)	276	292	6	74	67	_
Digestive system (C15–C21, C22–C25)	2478	582	−77	95	91	_
Respiratory system (C33–C39)	186	223	20	87	94	_
Melanoma and non-melanoma skin (C43, C44)	56	49	_	33	25	_
Breast (C50)	1025	1414	38	405	418	3
Gynecological (C51–C56)	574	498	−13	242	184	-24
Prostate (C61)	264	342	30	195	181	-7
Summary	4859	3400	−30	1131	1060	-6

* According to the International Classification of Disease, Tenth Revision (ICD-10).

**Table 3 healthcare-09-01522-t003:** The number of newly diagnosed cancer patients treated by surgery.

Solid Tumors	Year	Stage						Summary	Change, %
		I	II	III	IV	*p*-Value for ch2	*p*-Value for rsANOVA		
Colon (C18 *)	2019	15	40	47	25	0.099	0.32	127	−18
	2020	8	30	31	35			104	
Stomach (C16)	2019	36	35	70	38	0.56	0.046	179	−18
	2020	32	20	62	33			147	
Prostate (C61)	2019	36	234	82	20	0.67	>0.99	372	3
	2020	39	250	80	14			383	
Bladder (C67)	2019	88	28	4	6	0.64	>0.99	126	−15
	2020	71	22	6	8			107	
Kidney (C64)	2019	37	1	34	7	0.52	0.56	79	_
	2020	40	1	26	3			70	
Breast (C50)	2019	313	233	99	9	0.38	0.046	654	−8
	2020	281	229	91	3			604	
Lung (C34)	2019	33	28	90	83	0.053	0.046	234	−36
	2020	16	9	53	71			149	
Thyroid (C73)	2019	16	4	1	2	0.45	0.56	23	_
	2020	22	1	1	3			27	
Melanoma (C43)	2019	66	27	30	4	0.20	0.56	127	−1
	2020	51	38	30	7			126	
Larynx (C32)	2019	3	7	25	25	0.24	>0.99	60	_
	2020	7	6	17	33			63	
Summary	2019	643	637	482	219	0.36	0.046	1981	−10
	2020	567	606	397	210			1780	

* According to the International Classification of Disease, Tenth Revision (ICD-10).

## Data Availability

The data presented in this study are available on request from the corresponding author. The data are not publicly available due to ethical issues.

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
