# Peer review of "The Impact of the COVID-19 Pandemic on Cancer Patient’s Management—Lithuanian Cancer Center Experience"

_healthcare, 2021, doi:10.3390/healthcare9111522_

Round 1

Reviewer 1 Report

The manuscript deals with COVID-19 induced changes in cancer care during the first wave of infections. Overall, the Topic is important despite various preliminary reports which were mainly driven single Providers. Though, the national perspective for Lithuania can provide novel imformation. Various remarks seem to be required.

  • Since case numbers usually Show variations between the years it is recommended to compare 2020 with the same period in a multi-year view. eg. comparing the mean of 2017-2019
  • The statistical comparison should be done for entire year 2020, but also for subperiods (monthly of quarterly) - at least for some key numbers
  • The increase in psychological care is surprising and in contrast to other reports, is this the demand or real provision of services? Data should be added as for the other Treatment modalities.
  • Do the authors have data to compare with overall performance of healthcare? At least in the discussion this would be of interest. Various reports suggest that oncology had less reduction than other healthcare sectors. This might be due to fast adaptation of patient pathways (eg.  Clin Exp Metastasis 38, 257–261 (2021)

Author Response

1.Since case numbers usually Show variations between the years it is recommended to compare 2020 with the same period in a multi-year view. eg. comparing the mean of 2017-2019

Answer: Electronic Database (Hospital Information System) was introduced at National Cancer Institute only  in 2019, therefore to collect data from paper source for 2017-2018 years would be time consuming and collected data itself would be unreliable due to possible human error.

  1. The statistical comparison should be done for entire year 2020, but also for subperiods (monthly of quarterly) - at least for some key numbers

Answer: We took into account the data for the month in the periods March-May, June-August, September-December and compared the frequencies available in these periods by the chi-square method. Also, we included the calculated frequency change as a percentage in the text. These changes significantly strengthened the clarity and validity of the statements. The text was improved as follows:

Systemic anticancer treatment (SACT) is one of the core activities of the NCI with no significant changes in the extent of these services registered during the pandemic (Table 1). Regarding the summarized numbers of out-patient (day care) and in-patient units (Figure 1), time-dependent variations also had no a declining trend in 2020 vs 2019 (rsANOVA p=0.74). However, significant changes in frequency comparing three periods: March-May (-8%), June-August (-1%), and September-December (+9%) (chi2=58, df=2, p<0.001), revealed that SACT services decreased at the beginning of the quarantine, but later recovered and even increased.

Descriptive statistics revealed trends in time-dependent decrease of patient flow in 2020   compared to 2019 (rsANOVA p=0.13; Figure 2), and the drops in cases during March-May (-15%), June-August (-12%) and September-December (-5%) were comparable (ch2=2.15, df=2, p=0.34).

  1. The increase in psychological care is surprising and in contrast to other reports, is this the demand or real provision of services? Data should be added as for the other Treatment modalities.

Answer: According to the The National Audit Office of Lithuania, during the pandemic, about 10 % more people turned to medical institutions for depression, anxiety and reaction to severe stress disorders and the number of visits to primary mental health care professionals increased by about 17 % compared to 2019.

Our data showed the increase in mental health services for  cancer patients. Seeking to overcome hindrances to psychiatric and psychological services due to quarantine restrictions, we adopted telepsychiatry for cancer patients with emergency psychological problems. With the help of remote means of mental healthcare, a larger number of patients were counseled compared to 2019. The techniques of telepsychiatry for cancer patients requires further examination and refinement so that they may be fully utilized within the mental health service system.

  1. Do the authors have data to compare with overall performance of healthcare? At least in the discussion this would be of interest. Various reports suggest that oncology had less reduction than other healthcare sectors. This might be due to fast adaptation of patient pathways (eg.  Clin Exp Metastasis 38, 257–261 (2021).

Answer: COVID-19 pandemic and national lockdown negatively affected Lithuanian healthcare system. According to annual reports and compulsory health insurance database data provided by Health Information Centre of Institute of Hygiene [9] visits to physicians in Lithuania decreased by 34,4%, visits to oncologists decreased by 20,2% accordingly. The same trend was shown in our study. Pandemic and national lockdown had negative impact on majority of cancer diagnostic and treatment-related services provided by the NCI of Lithuania.

Reviewer 2 Report

The study titled "THE IMPACT OF COVID-19 PANDEMIC ON CANCER PATIENT’S MANAGEMENT – LITHUANIAN CANCER CENTRE EXPERIENCE" analyses the impact of COVID-19 on the cancer patients and their treatment. The authors analyse the data for two years 2019 and 2020 to assess the patient admission diagnosis and treatment at NCI, Lithuania. Such studies are crucial to determine the impact of pandemic or such emergencies on vulnerable sections of society and design SOPs or policies to avoid neglect of adequate healthcare provisions.

Major comments:

  1. The cancer patient screening data not shown, may be added to the manuscript. It will inform the extent of impact of COVID-19.
  2. Figure 3 data needs to be better represented. Percentages must be calculated for the respective year and not for the total cancer type to give better comparison. Vertical bar graph may be a better depiction of such data.

Author Response

The cancer patient screening data not shown, may be added to the manuscript. It will inform the extent of impact of COVID-19.

Answer: we expand Table 1 (please see the text).

Figure 3 data needs to be better represented. Percentages must be calculated for the respective year and not for the total cancer type to give better comparison. Vertical bar graph may be a better depiction of such data.

Answer: We have done changes according to your suggestions (please see the text).

Round 2

Reviewer 1 Report

all comments were sufficiently adressed

This manuscript is a resubmission of an earlier submission. The following is a list of the peer review reports and author responses from that submission.